# Quality of Life in Danish Patients with Multiple Myeloma during the COVID-19 Pandemic

Louise Redder [1,*], Sören Möller [2,3], Anna Thit Johnsen [4], Mary Jarden [5], Christen Lykkegaard Andersen [5], Bo Amdi Jensen [6], Henrik Frederiksen [1], Henrik Gregersen [7], Anja Klostergaard [8], Morten Saaby Steffensen [9], Per Trøllund Pedersen [10], Maja Hinge [11], Mikael Frederiksen [12], Carsten Helleberg [13], Anne Kærsgaard Mylin [5], Niels Abildgaard [1] and Lene Kongsgaard Nielsen [1,14]

1   Quality of Life Research Center, Department of Haematology, Odense University Hospital, 5000 Odense C, Denmark; Henrik.Frederiksen@rsyd.dk (H.F.); Niels.Abildgaard@rsyd.dk (N.A.); Lene.Kongsgaard.Nielsen@rsyd.dk (L.K.N.)
2   OPEN, Open Patient Data Explorative Network, Odense University Hospital, 5000 Odense C, Denmark; Soren.Moller@rsyd.dk
3   Department of Clinical Research, University of Southern Denmark, 5000 Odense C, Denmark
4   Department of Psychology, University of Southern Denmark, 5230 Odense M, Denmark; atjohnsen@health.sdu.dk
5   Department of Haematology, Copenhagen University Hospital, 2100 Copenhagen, Denmark; Mary.Jarden@regionh.dk (M.J.); christenla@gmail.com (C.L.A.); anne.kaersgaard.mylin@regionh.dk (A.K.M.)
6   Department of Haematology, Zealand University Hospital, 4000 Roskilde, Denmark; boaj@regionsjaelland.dk
7   Department of Haematology, Aalborg University Hospital, 9000 Aalborg, Denmark; henrik.gregersen@rn.dk
8   Department of Haematology, Aarhus University Hospital, 8200 Aarhus N, Denmark; Anja.Klostergaard@rm.dk
9   Department of Haematology, Regional Hospital West Jutland, 7500 Holstebro, Denmark; morten.saaby@rm.dk
10  Department of Haematology, South West Jutland Hospital, 6700 Esbjerg, Denmark; Per.Troellund.Pedersen@rsyd.dk
11  Department of Haematology, Vejle Hospital, 7100 Vejle, Denmark; Maja.Hinge@rsyd.dk
12  Department of Haematology, Hospital of Southern Jutland, 6200 Aabenraa, Denmark; Mikael.Frederiksen1@rsyd.dk
13  Department of Haematology, Herlev Hospital, 2730 Herlev, Denmark; carsten.helleberg@regionh.dk
14  Research Unit for Multimorbidity, Department of Internal Medicine and Cardiology, Viborg Regional Hospital, 8800 Viborg, Denmark
*   Correspondence: Louise.redder3@rsyd.dk; Tel.: +45-24-45-97-70

**Abstract:** In general, governments and health authorities have taken precautions during the COVID-19 pandemic to reduce the viral spread and protect vulnerable citizens. Patients with multiple myeloma (MM) have an increased risk of being infected with COVID-19 and developing a fatal course due to the related immunodeficiency. We investigated how Danish patients with MM reported their quality of life (QoL) pre-COVID and during COVID, in an ongoing longitudinal QoL survey. The responses given during the first and second wave of the COVID-19 pandemic were pooled, analyzed and compared to the same period the year before. We hypothesized that locking down the society would have caused deteriorated QoL and that patients living alone and those under the age of 65 would be particularly affected by the situation. Surprisingly, our study showed the opposite. Statistically significant and clinically relevant differences were primarily found during the first lock down and represented reduced fatigue, improved role functioning, decreased insomnia and improved physical health summaries in patients below 65 years of age. These results indicate that Danish patients with MM might have felt protected and safe by COVID restrictions. Otherwise, the questionaries used in QoL-MM survey may not have been able to capture the impact of the COVID-19 pandemic. Importantly, this indicates that QoL survey data obtained in clinical studies, in countries with highly developed health-care systems using standard questionnaires during the pandemic, allow room for interpretation without being adjusted for the impacts of the pandemic.

**Keywords:** multiple myeloma; quality of life; COVID-19 pandemic 4; EORTC QLQ-C30 5; EORTC QLQ-MY20 6; EORTC QLQ-CIPN20 7; SF12v2

## 1. Introduction

The severe acute respiratory syndrome coronavirus 2 (SARS-CoV-2) leading to coronavirus-19 (COVID-19), was detected for the first time in Wuhan, China, in December 2019 [1]. COVID-19 is currently raging globally causing millions of deaths, general lockdowns and social isolation.

Quarantine and isolation appear to be associated with poorer mental health such as post-traumatic stress symptoms, avoidance behaviors, depression, fear of own health and socioeconomic distress [2]. Certain groups have shown to be more prone to suffer from traumatic experiences due to the COVID-19 pandemic, e.g., women, younger and single adults, unemployed individuals who do not have others to care for them and people worried about financial instability and social isolation [3–5]. By contrast, older age and social support seem protective against depression, posttraumatic stress disorder (PTSD), anxiety disorder and suicidal ideation [5].

In patients with cancer, the COVID-19 pandemic and corresponding lockdown have resulted in impaired sleep, anxiety, pain, decrease in fitness, deterioration in emotional and social function and the less likelihood of contacting physicians regarding health-related concerns [6–10].

In general, and before the pandemic, patients living with multiple myeloma (MM) report impaired quality of life (QoL) compared to other cancer patients [11–13]. They experience reduced social functioning and work-life and have difficulties maintaining contact with the labor market [14,15]. Immunodeficiency caused by MM and its treatment cause increased risk of infection [16], and in general, patients with hematological malignancies including MM are at high risk of being infected with COVID-19 and experiencing a fatal outcome [17,18].

The difficulties caused by the COVID-19 pandemic in the general population could be comparable to everyday life with MM. Many patients with MM experience socioeconomic consequences from living with the disease, and they are familiar with taking special precautions due to their immunodeficiency. Thus, the COVID-19 pandemic might be particularly burdensome to patients living with MM causing noxious consequences in this challenged and vulnerable population. On the other hand, patients with MM are mostly part of the aging population with a mean age of 70 at the time of diagnosis [19]. Since social support protects the elderly against the psychological consequences of the COVID-19 pandemic [5], QoL might only be affected among the youngest and those living on their own.

We aimed to investigate the QoL of Danish patients with MM during the COVID-19 pandemic. We hypothesized that patients living alone and those under the age of 65 years as a consequence of the pandemic would have impaired QoL due to social isolation and fear of infection with SARS-CoV-2.

## 2. Materials and Methods

### 2.1. Study Design and Population

The Danish prospective, nation-wide, observational survey (QoL-MM) [20] framed our study. The study population was a subpopulation of the QoL-MM cohort referred to as QoL-MM-C19. The QoL-MM-C19 cohort was constructed based on the response time of the questionnaires, and QoL was compared using patient-reported outcome (PRO) data obtained before and during the COVID-19 pandemic on group level.

Participants eligible for inclusion in QoL-MM were newly diagnosed or relapsed, treatment-demanding patients with MM. Patients were excluded if they had primary refractive MM, were not able to understand Danish or had a psychiatric diagnosis or mental difficulties that prevented them from answering the questionnaires. All departments of hematology in Denmark recruited patients to the QoL-MM study in collaboration with the Danish Myeloma Study Group. The treating doctor or nurse introduced eligible participants to the QoL-MM study and all patients provided written consent before participating. Follow-up of patients ends after 24 months.

The first Danish citizens were officially diagnosed with COVID-19 at the end of February 2020, and the first general lockdown was a reality in Denmark in the middle of March 2020. The questionnaires completed during the first and second lockdown were compared to the questionnaires completed in the same time period one year before. Thus, the population for this study (the QoL-MM-C19 cohort) was composed based on the response time of the questionnaires. The following time periods were defined to represent the first and second wave of the pandemic and comparative control periods:

1.  April–June 2020 defines the first wave of the COVID-19 pandemic;
2.  April–June 2019 equals the comparative time period of the first wave;
3.  November 2020–January 2021 defines the second wave of the pandemic;
4.  November 2019–January 2020 equals the comparative time period of the second wave.

Participants answering at least one questionary during these time periods constitute the QoL-MM-C19 cohort. Thus, each participant contributed between one and three set of questionnaires during at least one of these time periods, and it was possible to participate in more than one period depending on how long they had been in the QoL-MM study.

### 2.2. Questionnaires

The following PRO questionnaires were used; the cancer-generic instrument of European Organisation for Research and Treatment of Cancer Quality of life (EORTC) QLQ-C30 (EORTC QLQ-C30), the Multiple Myeloma module QLQ-MY20 (EORTC QLQ-MY20), the Chemotherapy-Induced Peripheral Neuropathy module (EORTC QLQ-CIPN20) and the short-form health survey version 2 (SF12v2). The EORTC questionnaires use a 7-day recall period, whereas the SF12v2 questionnaire uses a 4-week recall period:

*   EORTC QLQ-C30 consists of five functional domains (physical, role, emotional, cognitive and social), nine symptom domains (fatigue, nausea and vomiting, pain, dyspnoea, insomnia, appetite loss, constipation, diarrhoea and financial difficulties) and one global health status/QoL [21,22].
*   EORTC QLQ-MY20 consists of two functional domains (future perspectives and body image) and two symptom scales (disease symptoms and side effects of treatment) [23].
*   EORTC QLQ-CIPN20 consists of three symptoms and problems of neuropathy (sensory, motoric and autonomic) [24] and the 18-item sum score, which is a multi-item domain excluding item 19 and 20 [25]. Here, the sum score will be presented.
*   SF12v2 consists of two health summaries (physical and mental) [26].

The questionnaires could be completed either online or via paper and pencil. Survey data were obtained at enrolment and subsequently at 12 follow-up time points over a two-year period, and, at each time point, 2–4 different PRO questionnaires were completed.

### 2.3. Statistical Analysis

We compared the QoL reported by the QoL-MM-C19 cohort during the first and second wave to the same time period a year before, e.g., the QoL of the QoL-MM-C19 cohort in April–June 2019 were compared to the cohorts' QoL in April–June 2020 (first wave of the COVID-19 pandemic). Groups were compared in a combined longitudinal model. Appropriate statistical power was ensured by including all relevant questionnaires in an overall model taking repeated questionnaires from the same individual into account. Scale scores for each patient were calculated in accordance with the related scoring manuals [27,28]. Between-group domain score differences were tested for statistical significance using mixed-effects linear regression, with a month-period interaction (e.g., April–June 2019 vs. April–June 2020), taking into account answers from the same patient as a random intercept.

A *p*-value of <0.05 was considered as statistically significant. Between-group differences (first/second wave vs. same period one year before) were considered clinically relevant if they reached the thresholds for a minimal important difference (MID) in accordance with Cohen's MID criteria (defined as 0.3 standard deviation of the mean score of

each domain for the entire group) [29,30]. A mean score difference was considered evident, if the difference between the comparative time period and first/second wave scale score were both statistically significant and clinically relevant. The analyses were repeated with questionnaire time point (e.g., 9 months follow-up questionnaire) included as a categorical covariate.

Patients under 65 years of age and those living alone (unmarried, separated, divorced or widows) were analyzed in two separate groups to investigate whether living alone or being younger were predictors of impaired QoL during the COVID-19 pandemic. Mixed-effects linear regressions were performed including all the answers given during first/second wave vs. the comparative period one year before, including a random intercept for each patient to take into account repeated questionnaire answers from the same patient.

In the baseline characteristics, Wilcoxon rank sum test was used to test for statistical difference in age between the periods in the QoL-MM-C19 cohort, and chi-squared test was used to test for differences in the categorical variables.

Stata statistical software was used for all statistical analyses.

### 2.4. Validation of the Dataset

Participants answering questionnaires in the corresponding months to the first and second wave in 2018 were used for validation to assure consistency of the dataset, e.g., QoL data of April–June 2018 were compared to those of April–June 2019. By comparing QoL and baseline characteristics in the corresponding two periods in 2018, the variation caused by, e.g., differences in inclusion flow or time in QoL-MM, were sought to be covered.

## 3. Results

### 3.1. Study Population

A total of 616 patients were included in QoL-MM by the end of January 2021. Females represented 41% of the population and the mean age was 68.6 years. No significant differences were found between cohorts regarding age, sex, marital status, IMWG frailty score, Karnofsky performance status, Charlson comorbidity index, Freiburg comorbidity index, alcohol consumption and smoking habits.

During the first wave, 389 questionnaires were completed and compared to 472 questionnaires completed during the same time period a year before. During the second wave, 349 questionnaires were completed and compared to 475 completed questionnaires a year before. The rate of non-response were between 3 and 4%. The mean time of participation in the QoL-MM study was 308.6 days during the first wave and 289.6 days during the second wave. During April–June 2019, the mean participation time was 266.9 days leading to a minor difference when compared to the first wave. Concerning the second wave, this difference was fully balanced; see Table 1.

The clinical status varied during the first and second wave and the comparative periods. This difference can be explained by coincidence, and, since RMM reports better QoL compared to NDMM [31], this difference is of less concern; see Table 1 and Table S1.

**Table 1.** Inclusion and questionnaire completion rates during the two periods.

| | Baseline Characteristic | | | | | |
|---|---|---|---|---|---|---|
| | **First Wave** | | | **Second Wave** | | |
| | **April–June 2020** | **April–June 2019** | ***p*-Value** | **November 2020–January 2021** | **November 2019–January 2020** | ***p*-Value** |
| Numbers of patients | 286 | 339 | | 249 | 341 | |
| Scheduled questionnaires | 399 | 494 | | 364 | 496 | |
| Mean time in QoL-MM Days (SD) | 308.6 (234.1) | 266.9 (214.7) | 0.004 | 289.6 (243.3) | 282.9 (249.5) | 0.319 |
| Mean Age (SD) | 68.6 (9.5) | 67.7 (9.1) | 0.153 | 68.3 (8.9) | 68.0 (9.3) | 0.823 |
| **Response to questionnaire invitation** | | | 0.120 | | | 0.935 |
| Responders N (%) | 389 (97%) | 472 (96%) | | 349 (96%) | 475 (96%) | |
| Non-responders N (%) | 10 (3%) | 22 (4%) | | 15 (4%) | 21 (4%) | |
| Sex N(%): | | | 0.748 | | | 0.672 |
| Female | 117 (41%) | 143 (42%) | | 103 (41%) | 147 (43%) | |
| Male | 169 (59%) | 196 (58%) | | 146 (59%) | 194 (57%) | |
| **Clinical status** | | | 0.288 | | | 0.024 |
| Newly diagnosed MM, N (%) | 290 (73%) | 343 (69%) | | 228 (63%) | 347 (70%) | |
| Relapse MM, N (%) | 109 (27%) | 151 (31%) | | 136 (37%) | 149 (30%) | |
| **Marital status** | | | 0.707 | | | 0.900 |
| Married/cohabiting N (%) | 218 (76%) | 254 (75%) | | 188 (76%) | 259 (76%) | |
| Single N (%) | 68 (24%) | 85 (25%) | | 61 (24%) | 82 (24%) | |
| **Alcohol units pr. week** | | | 0.871 | | | 0.859 |
| No | 58 (20%) | 70 (21%) | | 52 (21%) | 68 (20%) | |
| 1–7 | 153 (53%) | 188 (55%) | | 142 (57%) | 188 (55%) | |
| 8–14 | 51 (18%) | 52 (15%) | | 36 (14%) | 58 (17%) | |
| >14 | 24 (8%) | 29 (9%) | | 19 (8%) | 27 (8%) | |
| **Smoking:** | | | 0.149 | | | 0.635 |
| Current | 25 (9%) | 38 (11%) | | 20 (8%) | 34 (10%) | |
| Former | 150 (53%) | 152 (45%) | | 131 (53%) | 169 (50%) | |
| Never | 110 (39%) | 148 (44%) | | 97 (39%) | 137 (40%) | |
| **Method of answering N (%):** | | | 0.812 | | | 0.191 |
| Electronic by mail | 241 (84%) | 288 (85%) | | 221 (89%) | 290 (85%) | |
| Paper and pencil | 45 (16%) | 51 (15%) | | 28 (11%) | 51 (15%) | |
| **IMWG frailty score N (%):** | | | 0.458 | | | 0.695 |
| Fit | 146 (51%) | 187 (55%) | | 130 (52%) | 187 (55%) | |
| Intermediate Fitness | 96 (34%) | 110 (32%) | | 82 (33%) | 111 (33%) | |
| Frail | 44 (15%) | 42 (12%) | | 37 (15%) | 43 (13%) | |
| **Karnofsky performance status N (%)** | | | 0.229 | | | 0.778 |
| 100% | 111 (39%) | 113 (33%) | | 96 (39%) | 122 (36%) | |
| 90% | 99 (35%) | 132 (39%) | | 91 (37%) | 133 (39%) | |
| 80% | 31 (11%) | 49 (14%) | | 25 (10%) | 40 (12%) | |
| <=70% | 45 (16%) | 45 (13%) | | 37 (15%) | 46 (13%) | |
| **Charlson comorbidity index** | | | 0.944 | | | 0.724 |
| 0 | 151 (53%) | 187 (55%) | | 127 (51%) | 186 (55%) | |

**Table 1.** *Cont*.

| | Baseline Characteristic | | | | | |
|---|---|---|---|---|---|---|
| | First Wave | | | Second Wave | | |
| | April–June 2020 | April–June 2019 | *p*-Value | November 2020–January 2021 | November 2019–January 2020 | *p*-Value |
| 1 | 51 (18%) | 56 (17%) | | 41 (16%) | 59 (17%) | |
| 2 | 52 (18%) | 59 (17%) | | 51 (20%) | 60 (18%) | |
| 3+ | 32 (11%) | 37 (11%) | | 30 (12%) | 36 (11%) | |
| **Freiburg comorbidity index** | | | 0.674 | | | 0.977 |
| 0 | 223 (78%) | 268 (79%) | | 197 (79%) | 272 (80%) | |
| 1 | 56 (20%) | 66 (19%) | | 47 (19%) | 62 (18%) | |
| 2 | 7 (2%) | 5 (1%) | | 5 (2%) | 7 (2%) | |

SD; standard deviation, MM; multiple myeloma, IMWG: International Myeloma Working Group.

### 3.2. The QoL Reported by the QoL-MM-C19 Cohort

The QoL-MM-C19 cohort reported statistical significant reduced fatigue (*p*-value 0.03), reduced diarrhoea (*p*-value 0.016) and deterioration of mental health (*p*-value 0.007) during the first wave of the COVID-19 pandemic compared to the same period in 2019. None of these findings were clinically relevant, as the threshold of MID was not reached. No statistically significant differences were reported during the second wave of the COVID-19 pandemic when comparing the score in November 2020–January 2021 to November 2019–January 2020; see Table 2.

**Table 2.** Quality of life reported by the QoL-MM-C19 cohort.

| All Patients Included in the COVID-19 Cohort | | | | | | |
|---|---|---|---|---|---|---|
| | First Wave | | | Second Wave | | |
| | April–June 2020 Q = 389 | April–June 2019 Q = 472 | *p*-Value | November 2020–January 2021 Q = 349 | November 2019–January 2020 Q = 475 | *p*-Value |
| **EORTC-QLQ-C30 Mean (SD)** | | | | | | |
| Global health status QoL | 60.37 (22.56) | 61.04 (22.83) | 0.921 | 59.83 (21.46) | 60.52 (22.81) | 0.673 |
| **Functional scales** | | | | | | |
| Physical functioning | 72.47 (21.54) | 72.26 (20.48) | 0.067 | 70.81 (22.44) | 70.11 (22.65) | 0.670 |
| Role functioning | 61.44 (31.18) | 60.32 (28.93) | 0.087 | 60.32 (30.90) | 58.42 (32.54) | 0.300 |
| Emotional functioning | 79.91 (19.84) | 81.82 (18.90) | 0.255 | 80.21 (17.92) | 79.47 (20.80) | 0.802 |
| Cognitive functioning | 84.36 (19.53) | 82.91 (20.44) | 0.217 | 82.66 (20.17) | 81.33 (22.63) | 0.424 |
| Social functioning | 75.28 (28.05) | 76.79 (24.65) | 0.519 | 73.97 (27.30) | 76.79 (26.22) | 0.071 |
| **Symptom scales** | | | | | | |
| Fatigue | 38.43 (24.82) | 40.19 (24.31) | 0.030 | 39.80 (23.51) | 40.05 (25.48) | 0.768 |
| Nausea and vomiting | 6.86 (13.44) | 6.75 (14.43) | 0.966 | 7.14 (15.53) | 7.24 (14.55) | 0.802 |
| Pain | 29.01 (27.69) | 27.49 (26.76) | 0.356 | 30.56 (27.46) | 31.54 (29.66) | 0.443 |
| Dyspnoea | 19.19 (24.35) | 21.39 (26.05) | 0.055 | 23.88 (27.04) | 22.46 (27.03) | 0.431 |
| Insomnia | 24.68 (25.48) | 26.87 (28.47) | 0.899 | 29.61 (29.50) | 27.50 (29.34) | 0.605 |
| Appetite loss | 18.81 (28.22) | 15.82 (26.36) | 0.265 | 15.13 (24.81) | 16.60 (25.30) | 0.555 |
| Constipation | 15.38 (24.70) | 14.57 (23.03) | 0.689 | 18.30 (24.81) | 18.46 (27.05) | 0.498 |
| Diarrhoea | 12.17 (21.81) | 15.31 (24.79) | 0.016 | 15.80 (26.22) | 14.76 (23.52) | 0.742 |
| Financial difficulties | 7.03 (17.64) | 6.18 (17.53) | 0.790 | 7.28 (18.72) | 8.39 (19.23) | 0.276 |

**Table 2.** *Cont.*

| | **All Patients Included in the COVID-19 Cohort** | | | | | |
|---|---|---|---|---|---|---|
| | **First Wave** | | | **Second Wave** | | |
| | **April–June 2020** **Q = 389** | **April–June 2019** **Q = 472** | ***p*-Value** | **November 2020–January 2021** **Q = 349** | **November 2019–January 2020** **Q = 475** | ***p*-Value** |
| **SF 12v2 Mean (SD)** | | | | | | |
| Mental health summaries | 43.95 (10.96) | 46.54 (10.01) | 0.007 | 43.49 (11.29) | 44.68 (11.40) | 0.378 |
| Physical health summaries | 41.80 (10.27) | 41.50 (9.87) | 0.691 | 42.83 (10.10) | 41.09 (10.64) | 0.059 |
| **EORTC CIPN20 Mean (SD)** | | | | | | |
| Sum score | 13.52 (12.50) | 14.22 (12.92) | 0.570 | 14.49 (12.71) | 12.69 (12.90) | 0.149 |
| **EORTC QLQ-MY20 Mean (SD)** | | | | | | |
| Future perspectives | 66.48 (23.80) | 66.14 (22.72) | 0.476 | 61.78 (25.81) | 64.22 (25.78) | 0.272 |
| Body image | 76.95 (28.96) | 76.71 (27.93) | 0.876 | 75.70 (26.57) | 76.78 (28.76) | 0.551 |
| Disease symptoms | 21.45 (17.89) | 21.67 (18.77) | 0.947 | 23.43 (18.80) | 25.22 (18.94) | 0.681 |
| Side effect of treatment | 19.80 (15.67) | 17.73 (13.83) | 0.065 | 18.23 (14.74) | 18.21 (14.17) | 0.974 |

Neither of the differences are statistically significant and clinically relevant. Q; questionnaires, SD; standard deviation, QoL; quality of life.

### 3.3. Subgroup Analysis, QoL among Patients Living Alone and Patients under 65 Years

The group of patients living alone reported improved role functioning during the first wave reaching both statistical significance (*p*-value <0.001) and the threshold of MID. Fatigue and physical function were statistically significantly improved (*p*-value 0.027 and 0.030) and the sum score of neuropathy was increased (*p*-value 0.049), but the thresholds of MID were not reached. Patients living alone reported no statistically significant differences during the second wave, and the improvement seen during the first wave could no longer be detected (Table 3).

Patients under 65 years reported improved physical health summaries (*p*-value 0.016), decreased fatigue (*p*-value < 0.001), less insomnia (*p*-value 0.002) and improved role functioning (*p*-value <0.001) during the first wave, reaching both statistical significance and the threshold of MID. They also reported statistically significant improvement of diarrhea (*p*-value 0.018), dyspnea (*p*-value 0.004), pain (*p*-value 0.009), physical functioning (*p*-value 0.001), global health status QoL (*p*-value 0.045) sum neuropathy score (*p*-value 0.014) and disease symptoms (*p*-value 0.025), but none of these domains reached the threshold of MID.

During the second wave, the patients under the age of 65 reported statistically significant improved physical health summaries (*p*-value 0.004) also reaching the threshold of MID. No other domains showed statistical differences during the second wave (Table 4).

### 3.4. Validation

To validate the analyses and to investigate whether significant differences were to be expected when comparing QoL on group level one year apart, we investigated the consistency of the QoL data by analyzing the data for the study periods in 2018. When comparing the answers to questionnaires completed in April–June 2018 to the questionnaires completed in April–June 2019 and comparing the questionnaires completed in November 2018–January 2019 to the questionnaires completed in November 2019–January 2020, none of the differences were statistically significant and clinically relevant.

Baseline characteristics and scores of all 25 domains investigated in QoL-MM during the 6 time periods (April–June in 2018, 2019 and 2020 and the following November–January) are presented in the Supplementary Tables S1 and S2. They include the QoL-MM-C19 cohort and the subpopulations of participants living alone and patients under the age of 65 years.

**Table 3.** Quality of life reported by the subpopulation living alone in the QoL-MM-C19 cohort.

| | COVID-19 Cohort Patients Living Alone | | | | | |
|---|---|---|---|---|---|---|
| | First Wave | | | Second Wave | | |
| | April–June 2020 Q = 98 | April–June 2019 Q = 123 | *p*-Value | November 2020–January 2021 Q = 87 | November 2019–January 2020 Q = 117 | *p*-Value |
| **EORTC-QLQ-C30 Mean (SD)** | | | | | | |
| Global health status QoL | 63.89 (20.76) | 59.69 (24.96) | 0.316 | 59.62 (19.25) | 60.60 (23.91) | 0.706 |
| **Functional scales** | | | | | | |
| Physical functioning | 74.00 (21.17) | 70.61 (20.22) | 0.032 | 70.54 (22.13) | 67.96 (21.03) | 0.799 |
| Role functioning | 66.50 (30.69) | 56.15 (29.71) | **<0.001** | 61.69 (33.83) | 57.55 (32.50) | 0.662 |
| Emotional functioning | 83.50 (16.23) | 82.37 (20.16) | 0.975 | 80.97 (16.39) | 81.32 (18.71) | 0.455 |
| Cognitive functioning | 84.52 (18.42) | 80.62 (23.02) | 0.064 | 77.97 (21.94) | 79.63 (21.68) | 0.637 |
| Social functioning | 78.74 (28.70) | 77.82 (24.85) | 0.416 | 73.37 (30.34) | 75.50 (27.38) | 0.525 |
| **Symptom scales** | | | | | | |
| Fatigue | 35.03 (21.95) | 40.35 (25.96) | 0.027 | 41.99 (23.14) | 41.69 (25.48) | 0.938 |
| Nausea and vomiting | 7.31 (13.55) | 6.23 (15.45) | 0.624 | 9.00 (15.21) | 9.40 (16.14) | 0.975 |
| Pain | 22.96 (25.39) | 27.64 (28.93) | 0.536 | 30.84 (26.48) | 32.19 (30.62) | 0.694 |
| Dyspnoea | 19.73 (24.33) | 17.91 (21.96) | 0.695 | 27.20 (27.62) | 25.64 (26.76) | 0.256 |
| Insomnia | 23.47 (22.56) | 26.50 (29.98) | 0.713 | 29.50 (31.92) | 27.07 (29.01) | 0.823 |
| Appetite loss | 16.84 (25.96) | 16.80 (25.74) | 0.982 | 18.01 (28.67) | 19.94 (26.65) | 0.749 |
| Constipation | 12.59 (21.68) | 16.26 (24.65) | 0.556 | 17.62 (23.21) | 23.93 (27.97) | 0.285 |
| Diarrhoea | 10.88 (19.60) | 14.36 (23.79) | 0.253 | 14.56 (26.75) | 16.52 (25.75) | 0.287 |
| Financial difficulties | 10.54 (22.75) | 10.47 (23.58) | 0.973 | 13.79 (25.70) | 12.36 (22.21) | 0.793 |
| **SF 12v2 Mean (SD)** | | | | | | |
| Mental health summaries | 46.37 (10.20) | 46.53 (11.48) | 0.742 | 42.92 (11.79) | 44.74 (10.50) | 0.473 |
| Physical health summaries | 44.34 (8.64) | 41.10 (9.59) | 0.114 | 44.65 (10.17) | 42.56 (11.02) | 0.499 |
| **EORTC CIPN20 Mean (SD)** | | | | | | |
| Sum score | 10.88 (8.95) | 13.44 (12.08) | 0.103 | 14.77 (14.11) | 10.86 (10.63) | 0.049 |
| **EORTC QLQ-MY20 Mean (SD)** | | | | | | |
| Future perspectives | 70.19 (22.42) | 67.92 (24.62) | 0.616 | 64.86 (28.87) | 68.28 (24.05) | 0.222 |
| Body image | 79.67 (27.77) | 73.58 (30.91) | 0.303 | 77.48 (28.39) | 77.42 (30.64) | 0.823 |
| Disease symptoms | 20.73 (17.34) | 21.05 (17.71) | 0.647 | 24.18 (20.28) | 25.22 (18.82) | 0.923 |
| Side effect of treatment | 17.49 (14.03) | 18.84 (14.92) | 0.237 | 18.97 (16.77) | 16.44 (11.72) | 0.159 |

*p*-values that are both statistically significant and clinically relevant are marked in bold. Q; questionnaires, SD; standard deviation, QoL; quality of life.

**Table 4.** Quality of life reported by the subpopulation of patients under 65 years in the QoL-MM-C19 cohort.

| | COVID-19 Cohort Patients under 65 Years | | | | | |
|---|---|---|---|---|---|---|
| | First Wave | | | Second Wave | | |
| | April–June 2020 Q = 128 | April–June 2019 Q = 157 | *p*-Value | November 2020–January 2021 Q = 123 | November 2019–January 2020 Q = 154 | *p*-Value |
| **EORTC-QLQ-C30 Mean (SD)** | | | | | | |
| Global health status QoL | 62.07 (23.04) | 58.98 (24.57) | 0.045 | 60.00 (21.86) | 58.88 (22.71) | 0.567 |
| **Functional scales** | | | | | | |
| Physical functioning | 77.34 (19.91) | 72.27 (21.78) | 0.001 | 72.38 (24.15) | 73.98 (20.86) | 0.693 |

**Table 4.** *Cont.*

| | COVID-19 Cohort Patients under 65 Years | | | | | |
|---|---|---|---|---|---|---|
| | **First Wave** | | | **Second Wave** | | |
| | **April–June 2020 Q = 128** | **April–June 2019 Q = 157** | ***p*-Value** | **November 2020–January 2021 Q = 123** | **November 2019–January 2020 Q = 154** | ***p-*Value** |
| Role functioning | 63.02 (31.30) | 54.67 (30.65) | **<0.001** | 59.62 (32.36) | 57.25 (33.27) | 0.197 |
| Emotional functioning | 78.39 (19.41) | 78.37 (21.16) | 0.538 | 80.69 (15.62) | 75.05 (21.51) | 0.215 |
| Cognitive functioning | 83.20 (19.34) | 79.94 (23.32) | 0.064 | 81.17 (20.57) | 79.87 (24.15) | 0.530 |
| Social functioning | 70.18 (32.11) | 72.12 (26.72) | 0.909 | 68.70 (29.38) | 75.00 (27.23) | 0.112 |
| **Symptom scales** | | | | | | |
| Fatigue | 37.15 (26.51) | 43.17 (28.52) | **<0.001** | 39.11 (23.04) | 41.27 (25.65) | 0.309 |
| Nausea and vomiting | 8.98 (15.78) | 9.13 (17.14) | 0.491 | 8.94 (16.57) | 9.85 (17.42) | 0.911 |
| Pain | 26.04 (24.90) | 31.53 (26.46) | 0.009 | 31.84 (28.23) | 31.93 (28.39) | 0.351 |
| Dyspnoea | 14.84 (19.08) | 22.51 (25.66) | 0.004 | 21.14 (25.35) | 19.26 (24.63) | 0.778 |
| Insomnia | 22.92 (23.57) | 32.70 (30.54) | **0.002** | 31.98 (31.77) | 31.39 (29.08) | 0.979 |
| Appetite loss | 18.23 (28.63) | 15.92 (27.11) | 0.963 | 16.26 (26.44) | 15.37 (23.84) | 0.849 |
| Constipation | 10.50 (20.45) | 12.31 (22.74) | 0.615 | 16.80 (25.02) | 16.02 (25.05) | 0.822 |
| Diarrhoea | 13.80 (24.93) | 17.62 (26.57) | 0.018 | 16.53 (25.38) | 15.80 (23.55) | 0.744 |
| Financial difficulties | 13.54 (24.55) | 11.54 (24.73) | 0.644 | 10.93 (23.25) | 14.94 (25.58) | 0.125 |
| **SF 12v2 Mean (SD)** | | | | | | |
| Mental health summaries | 43.84 (11.12) | 45.09 (9.87) | 0.254 | 42.24 (11.14) | 44.63 (11.15) | 0.059 |
| Physical health summaries | 43.85 (8.46) | 40.69 (10.46) | **0.016** | 46.02 (9.89) | 42.33 (10.40) | **0.004** |
| **EORTC CIPN20 Mean (SD)** | | | | | | |
| Sum score | 9.60 (9.78) | 12.49 (12.13) | 0.014 | 11.92 (10.40) | 10.22 (11.76) | 0.867 |
| **EORTC QLQ-MY20 Mean (SD)** | | | | | | |
| Future perspectives | 65.43 (20.21) | 62.06 (22.34) | 0.938 | 60.52 (25.24) | 62.00 (25.39) | 0.887 |
| Body image | 73.46 (30.63) | 69.05 (29.66) | 0.521 | 75.00 (26.40) | 71.19 (32.38) | 0.632 |
| Disease symptoms | 19.09 (14.05) | 23.90 (17.46) | 0.025 | 21.93 (19.06) | 25.51 (19.50) | 0.307 |
| Side effect of treatment | 19.86 (14.54) | 19.96 (16.42) | 0.508 | 16.30 (12.93) | 16.86 (14.98) | 0.941 |

*p*-values that are both statistical significant and clinical relevant are marked in bold. Q; questionnaires, SD; standard deviation, QoL; quality of life.

## 4. Discussion

In this study, we investigated whether patients with MM as a group report impaired QoL during the first and second wave of the COVID-19 pandemic. In contrast to the reported QoL in studies of cancer patients during the COVID-19 pandemic [6–10], no deterioration was captured by the questionnaires used in the QoL-MM study. The QoL-MM-C19 cohort was in our opinion a vulnerable population, since they were all receiving active treatment when included in the QoL-MM study and thus very threatened by infections. We thought that further restrictions added to the patients' existing precautions would lead to deterioration in their QoL, but we observed improvements in few domains.

As the COVID-19 infection spread in Denmark, the government took precautions to protect the most vulnerable citizens by locking down the society. Clinical practice was also changed by the health authorities [32] to reduce the MM patients' risk of being infected with COVID-19. The increased risk of infections that patients with MM usually are dealing with can lead to psychosocial consequences such as feeling alone or isolated [33]. However, in this case, the precautions taken by the government and health authorities might not

only have protected their physical wellbeing but also the mental health of patients living with MM. The society was now taking care of citizens with increased risk of infections, and patients with MM might have experienced it as a relief, as they were no longer the only ones being careful not to be infected by others. The fact that stable and, for a few domains, improved QoL were captured in this study supports the idea that the participants in the QoL-MM-C19 cohort had felt cared for and felt that they were in good hands during the first and second wave of the COVID-19 pandemic.

A Danish population survey investigating the impact of the first COVID-19 lockdown on risk of stress/depression found a reduced risk of depression during the first lockdown [34]. This positive effect of the first lockdown correlates with our findings among the patients under 65 years. The lockdown might have caused a feeling of relief due to a slower-paced life resulting in improved role function, fatigue, insomnia and physical health. As in the Danish population survey, this effect was gone shortly after first lockdown, and role functioning, fatigue and insomnia were not influenced by the second wave.

Methodological issues in the setup of QoL-MM might also explain our findings. The QoL-MM-C19 cohort were answering the questionnaires as participants in a myeloma survey without any information or questions drawing their attention toward COVID-19. When answering the questionnaires, the participants might have excluded the influence of the COVID-19 pandemic on their QoL. e.g., when asked whether their physical condition or medical treatment interfered with their social activities, the majority might have concluded that MM was influencing their social activities less than the COVID-19 pandemic. Correspondingly, they might have answered "not at all", meaning not at all compared to the restrictions caused by the pandemic. As the instruments used in this study are developed and validated to measure cancer and myeloma symptoms and side effects to treatment, the answers are probably less sensitive toward capturing the effects of the pandemic. In future studies, the influence of events not related to the purpose of the questionaries could be investigated by interviewing the participants' mapping of how they take such events into account.

When looking at the QoL during the six defined time periods from April 2018 to January 2021, very few significant differences were found, and the differences being both statistically significant and clinically relevant were exclusively found during the first wave of COVID-19, which represented improvements. Minimal important difference (MID) was used to compensate for multiple testing. Thus, as a group, the QoL-MM-C19 cohort shows stable QoL over time and reports no negative affect of the COVID-19 pandemic. This aligns with the work by Tom Atkinson [35] and Michalos et al. [36] reporting stability of QoL over time in the general population. Changes in life circumstances do affect QoL, but the majority recover and return to the level of QoL they had prior to the changes. A subjective measure as PRO repeatedly assessed and compared over time captures both changes in the patients' perceived QoL and the patients' adaption to a new health situation. This phenomenon is well-described in the theory of response shift [37]. Patients living with MM might have adapted to the risk of infection before the COVID-19 pandemic and therefore being less affected by the pandemic than other populations.

It is important to emphasize that our findings describe the situation in Danish patients with MM and the situation might of course be different in other societies or in other patient populations.

## 5. Conclusions

Based on our study, the COVID-19 pandemic did not negatively affect the reported QoL in Danish patients with MM. This observation may partly be explained by the fact that the questionnaires used were not specifically developed to capture the impact of a pandemic on QoL, but more general and MM-associated QoL. Importantly, these results suggest that QoL assessed by standard questionnaires in clinical trials in countries with highly developed health-care systems during the pandemic can be interpreted without adjusting for the impact of the pandemic.

**Supplementary Materials:** The following are available online at https://www.mdpi.com/article/10.3390/covid1010024/s1, Table S1: Baseline characteristic in all 6 time period, Table S2: Quality of life data reported during all 6 time periods by the QoL-MM-C19 cohort and subpopulations.

**Author Contributions:** Conceptualization, L.R., N.A. and L.K.N.; methodology, L.R., L.K.N., N.A. and S.M.; software, S.M.; validation, S.M. and N.A.; formal analysis, S.M.; investigation, L.R. and L.K.N.; resources, C.L.A., H.G., A.K., M.S.S., P.T.P., M.H., M.F., B.A.J., C.H., A.K.M., N.A. and L.K.N.; data curation, L.R., S.M. and L.K.N.; writing—original draft preparation, L.R.; writing—review and editing, A.T.J., M.J., C.L.A., H.F., H.G., A.K., M.S.S., P.T.P., M.H., M.F., B.A.J., C.H., A.K.M., N.A. and L.K.N.; visualization, L.R. and S.M.; supervision, A.T.J., N.A. and L.K.N.; project administration, L.R.; funding acquisition, L.R., N.A. and L.K.N. All authors have read and agreed to the published version of the manuscript.

**Funding:** This study was funded by The Danish Cancer Society (Grant Number R150-A10023), The Faculty of Health Sciences at the University of Southern Denmark, The Region of Southern Denmark, Amgen A/S, Celgene A/S, Takeda A/S and Janssen A/S.

**Institutional Review Board Statement:** The study was approved by the Danish Data Protection Agency, registered at ClinicalTrials.gov by number NCT02892383 and carried out in accordance with the Helsinki Declaration and Good Clinical Practice guidelines.

**Informed Consent Statement:** All participants have given informed consent.

**Data Availability Statement:** Access to data is restricted according to the Danish Data Protection Agency and legal Danish Health Care authorities.

**Conflicts of Interest:** L.R., N.A. and L.K.N. declare unrestricted research grants from Amgen A/S, Celgene A/S, Takeda A/S and Janssen A/S. All other authors declare no conflict of interest.

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
