# Peer review of "Quality of Life in Danish Patients with Multiple Myeloma during the COVID-19 Pandemic"

_covid, doi:10.3390/covid1010024_

Round 1

Reviewer 1 Report

This is an interesting paper regarding the exploring QoL of MM patients during the pandemic. The article is well written however, the interpretation of the results and the final conclusions are not well documented. There is too much information which offers conflicting conclusions. Probably you need to validate the results in a prospective fashion using additional questionnaires which could capture the true impact of the pandemic.

Reviewer 2 Report

Patient quality of life has gained recently much attention during the prolonged treatment offered to cancer patients.

This is an interesting manuscript that investigates a quality of life of Scandinavian myeloma patients during two waves of the present COVID pandemy. The investigation is well designed, the controls and questionnaires are appropriate.

Their most important conclusion is that in the well-to-do and well organized Scandinavian society in Denmark, patients felt protected by the COVID lockdown and their quality of life -already damaged by myeloma- did not further deteriorate due to COVID threat.

Unfortunately, in other societies with less delveloped social network and less organized health care system, the results may have been opposite. Nevertheless, the study is sound and is important for the myeloma community.

Author Response

Thank you for reviewing the manuscript and important comments.

We indeed acknowledge that our findings could have been different if explored in other communities, which we have added to the manuscript,

Line 334-336: “It is important to emphasize that our findings describe the situation in Danish patients with MM and the situation might of course be different in other societies or in other patient populations.”

We have added the following to the conclusion:

Line 340: added “Danish” and “This observation may partly be explained by the fact

The following is added to the abstract:

Line 45: “in highly developed health-care countries